# "It's a mother's choice": Exploring personal experiences, community support, cultural influences, and breastfeeding alternatives in Florence, Italy

**Megan Morley** [1]*, **Anjali Natarajan**[2], **Nicole A. Stepp**[2], **Andrea L. DeMaria**[1,3]

**1** Department of Public Health, College of Health and Human Sciences, Purdue University, West Lafayette, Indiana, United States of America, **2** School of Health Sciences, College of Health and Human Sciences, Purdue University, West Lafayette, Indiana, United States of America, **3** March of Dimes, Arlington, Virginia, United States of America

* megan_e_morley@icloud.com

## Abstract

This study explores the complex interplay of personal attitudes, community dynamics, and societal influences on breastfeeding experiences among women in Florence, Italy. Twenty-four women, aged 25 to 62, participated in comprehensive, 60-minute, in-person interviews conducted in May and June 2023. Four central themes emerged: 1) personal experiences, opinions, and attitudes, 2) communities involved in breastfeeding, 3) public opinion and culture, and 4) breastfeeding alternatives. Participants shared both emotional challenges and benefits, such as bonding and health advantages, while navigating the practical and social complexities of breastfeeding. Community support, including input from healthcare providers, midwives, and family networks, was crucial but varied in quality, sometimes offering encouragement and at other times presenting challenges. The findings highlight the individualized nature of breastfeeding durations, shaped by each mother's unique motivations and societal expectations. This study emphasizes the need for tailored support, public education, and comprehensive resources to foster positive breastfeeding experiences. The research advocates for policy and programmatic efforts that recognize and respect the diverse needs of mothers, promoting a supportive environment amid cultural and societal pressures.

## Introduction

Despite the World Health Organization's (WHO) recommendation to exclusively breastfeed until 6 months and continue breastfeeding alongside solid food consumption until 2 years, breastfeeding rates remain low worldwide [1]. Rates of exclusive breastfeeding vary worldwide, from 20% of infants in Central and Eastern Europe to 44% of infants in Southern Asia being exclusively breastfed at six months [2]. Several societal aspects contribute to the low rates, including nutritional, psychosocial, lifestyle, and medical factors [3]. Individual factors, such as a negative perception of breastmilk production, mastitis, and poor infant health, also impact breastfeeding trends [3]. Additionally, the incidence of breastfeeding is strongly

**Data availability statement:** Data relevant to this study are available from the Purdue University Research Repository: https://purr.purdue.edu/publications/4726/1

**Funding:** This research was partially funded by the Purdue University Office of Programs for Study Abroad, International Programs (Study Abroad and International Learning Grant and Intercultural Pedagogy Grant to AD). The funders had no role in study design, data collection and analysis, decision to publish, or preparation of the manuscript. No additional external funding was received for this study.

**Competing interests:** The authors have declared that no competing interests exist.

impacted by prenatal education, and the continuity of this practice is influenced by prenatal and postpartum care [4]. Postpartum education within the first few hours after birth correlates positively with breastfeeding rates of infants at 6 months [4].

Universally, breastfeeding is considered the best option for infant feeding. The adequate nutrition provided through breastmilk positively influences an infant's development and helps to prevent later-onset diseases [5]. Breastfeeding critically impacts public health as it has psychological, economic, and environmental benefits [6]. These benefits are associated with a lower risk of gastrointestinal infections, ear infections, asthma, childhood obesity, respiratory disease, and sudden infant death syndrome [6]. Therefore, increasing breastfeeding practices and duration is considered a public health priority, as international communities, organizations, and scientists support the WHO recommendation [6].

## Breast feeding in Europe & Italy

Variation exists in data collection methodology, breastfeeding promotion, and breastfeeding rates among European countries, thus making interpretation and comparison of relevant European data difficult [7]. That being said, WHO has been able to partly collect breastfeeding data for the 53 countries in WHO's European region. From 2006 to 2012, only an estimated 25% of infants in these member states were exclusively breastfed for the first six months [8]. Research conducted in 2016 reports data on breastfeeding rates within the first hour after birth and exclusive breastfeeding during the first four and six months [6]. Of the data shared, twenty-one countries reported breastfeeding rates within one hour after birth ranging from 5-84% (median=43%). Thirteen countries reported exclusive breastfeeding rates during the first four months ranging from 6-66% (median=33%). Twenty-four countries reported exclusive breastfeeding rates during the first six months ranging from 2-56% (median=23%) [6]. The World Breastfeeding Trends Initiative (WBTi) [9] has taken action to obtain more current data to understand breastfeeding globally. WBTi reports have been obtained in Europe from eighteen countries, including Italy, with indicators for policy and programs and infant feeding practices. However, not all countries collect data concerning all researched breastfeeding indicators, further complicating interpretation [9]. Twelve European countries collected data on breastfeeding initiation within one hour of birth ranging from 21-84% (median=60.5%). Sixteen European countries reported exclusive breastfeeding during the first six months, ranging from 10-65% (median=25.5%) [9]. The comparison of these more recent WBTi data points with WHO data is difficult since they are collected from a smaller sample size. However, these large differences across countries may be attributed to different societal views on breastfeeding in public, which can affect the duration of breastfeeding [10].

In Italy, breastfeeding data varies widely across regions, making interpretation challenging due to inconsistent indicators and a lack of standardized methodology [11]. As of 2013, The Italian National Statistics Institute stated exclusive breastfeeding rates during the first month, during the first three months, and at six months as 48.7%, 43.9%, and 5.5%, respectively [12]. A 2022 report provided additional data, indicating that 46.7% of infants aged 2–3 months and 30% of those aged 4–5 months were exclusively breastfed [13]. These decreases in exclusive breastfeeding may result from a complicated interplay of women trying to conform to deeply ingrained motherhood pressures and shame from public breastfeeding [14]. Exclusive breastfeeding rates also varied significantly by region, with higher rates observed in the north-central regions and lower rates in the south. Tuscany reported the second-highest exclusive breastfeeding rate at 61% for infants at 3 months and the highest rate of continued breastfeeding at 49.7% for children at 12 months [13]. However, Tuscany did not collect data on exclusive breastfeeding between 3 and 12 months, highlighting a gap that calls for

further research. As the capital of Tuscany, Florence offers insights into factors that may drive regional variability—such as culture, community influences, and public opinion—affecting breastfeeding rates, duration, and practices.

## Study purpose

This study aimed to explore women's perceptions and behaviors surrounding breastfeeding in and around Florence, Italy, to fill a gap in breastfeeding research. This was done by investigating attitudes through in-depth interviews with women who gave birth in Italy. Findings provide a better understanding of shared experiences and general perceptions to improve maternal health services in Italy. Florence was chosen as the study site for its unique cultural and historical context, which shapes healthcare practices and attitudes toward maternal care. Despite this, the region remains underrepresented in current breastfeeding research.

## Materials and methods

### Ethics statement

This study was approved by the Purdue University Institutional Review Board (IRB-2022-1472) with a letter of support from Florence University of the Arts. The research conformed to all ethical principles for medical research on human subjects, per the Declaration of Helsinki.

### Study

This study was part of a wider project investigating pregnancy and postpartum experiences, including family planning, fertility, healthcare support, birth plan and execution, breastfeeding, body image, and pregnancy and postpartum workplace policies. Researchers recruited participants from May 19th through May 31st, 2023, and conducted 24 English-language interviews in May and June 2023. Eligibility criteria included women at least 18 years of age and older who have given birth in Italy. Women who did not meet these criteria or did not speak conversational English were not considered for the study. This study used qualitative methodology to provide insights and perspectives on the thoughts and experiences of the participants. A descriptive approach was chosen to closely align with participants' perspectives, offering an authentic account of their lived experiences with minimal interpretation. Although the semi-structured protocol included some close-ended questions, these primarily served as prompts to spark conversation and invite deeper reflection. This design allowed participants to expand on their responses naturally, ensuring that the data collected captured both structured input and spontaneous insights. The last author's university Institutional Review Board approved the study with support from the partnering Italian institution.

Participants were mainly enlisted through in-person recruitment. Snowball sampling was also used, in which participants were asked to recommend other eligible individuals for the study [15]. Communication between researchers and participants was primarily held through email, where contact information was kept separate from their participant identification number. Each interview was conducted in a relaxing and accessible location chosen by the participant and lasted between 22 to 73 minutes. Interviews were audio-recorded using otter.ai, which is a recording and transcription software. Once the interview was completed, participants filled out an anonymous demographic questionnaire and were offered 20 euros compensation for their time and participation.

Interviews were conducted using a semi-structured protocol, allowing the researchers flexibility to add or change questions. This enabled the researcher to ask probing questions so the participants could add insights relevant to the study and their experiences. The beginning of the interview focused on general questions about the participant's experience living in Italy

and general history to build rapport and create a comfortable environment [15]. Additionally, this allowed for a robust understanding of breastfeeding attitudes and experiences among participants. See Table 1 for the full list of probing questions related to the research question.

## Research team

Data were collected and transcribed by 24 undergraduate students as part of a research-focused study abroad program. Immersed in Florentine culture, these students received graduate-level qualitative research methods training. The first two authors, who were also among the original interviewers, handled coding and analysis for this paper, ensuring consistency and a deep familiarity with the data throughout the study. The Principal Investigator and Program Teaching Assistant supervised all procedures and outcomes to maintain data reliability. The authors employed code manuals and mind mapping to identify themes, and they met regularly to discuss coding and explore emerging themes. Discrepancies were resolved through group consensus.

## Data analysis

Data analysis was conducted by four undergraduate students with backgrounds in qualitative and quantitative research; this process was overseen by a professor with an extensive research background in interdisciplinary women's reproductive and sexual health. This study used the theoretical framework of thematic analysis. An immersive, full content review was conducted to ensure familiarity with all data, noting immediate patterns or ideas for potential codes and themes [16,17].

**Table 1. Interview Questions.**

| Primary Question | Probing Questions |
|---|---|
| Did you breastfeed? | [IF YES]<br>• Tell me about your experience breastfeeding.<br>• How did you learn to breastfeed? Who taught you to breastfeed? Someone at the hospital (lactation consultant)? Mother? Someone in your community?<br>• Did you ever struggle to breastfeed? What were your struggles? What feelings did this evoke? How did you overcome those struggles?<br>• Did you have access to a breast pump? Where did you get it from? Did you choose to pump?<br>• Did you ever have to supplement breastfeeding? (e.g., formula, milk bank) Tell me about this experience.<br>• Do you believe breastfeeding is "better" than other feeding options? If so, what benefits do you think breastfeeding has?<br>• When did you stop breastfeeding? Did you feel supported in this decision to stop?<br>[IF NO]<br>• What factors or barriers led to your decision not to breastfeed?<br>• Did you feel pressure to breastfeed? Who was responsible for this pressure-social media, family, friends, doctors?<br>• Did you ever feel negative about not breastfeeding your baby? Please tell me about this.<br>• How did you choose a formula? What were you looking for in a formula? |
| Describe the breast-feeding culture in your community. Is breastfeeding something most women do? | • How long do women typically breastfeed for?<br>• Do women breastfeed in public? Are there places for women to privately breastfeed in public spaces?<br>• Do work places typically accommodate breastfeeding women? How so? Policies? Private Spaces? Dedicated time to feed or pump?<br>• How often do you see women breastfeeding in public in Florence?<br>• What messages (commercials, advertisements, etc.) have you seen about breastfeeding in your community?<br>• Do women typically feel comfortable talking to each other about breastfeeding? |

Then, a codebook was developed using both an inductive and deductive approach to allow for full data representation. Initially, codes were generated through a deductive process to compile the preliminary codebook. An inductive process allowed emerging themes to be better captured by adding and/or modifying codes. HyperRESEARCH 4.5.1 was used to complete multiple rounds of coding to achieve full data saturation, meaning no additional codes were being added to the data set. The same four undergraduate students completed codebook development and coding.

After coding, the authors organized data into developed themes and subthemes. Theme development was data-driven and closely reflected participant responses [16,17]. The authors reviewed these developed themes through two stages to refine them to capture participants' voices best. However, it was not feasible to collect participant theme feedback because of the timeline and study abroad aspect of this research. To accurately develop themes to best show the essence of the data, authors thoroughly collaborated and discussed the main themes and then created various subthemes to provide structure and create new levels of deeper meaning. Any data discrepancies were discussed and reviewed until the final theme development was achieved.

### Research participants

The mean age of the 24 interview participants was 44.9 years (SD = 11.1, range = 25-62). All (n=24, 100%) participants resided in or around Florence at the time of the study, and all (n=24, 100%) participants indicated they were cisgender women. Most (n=22, 91.7%) participants were in a relationship with one partner, and two (8.3%) were single. Two (8.3%) participants completed high school, twelve (50%) completed undergraduate education, and ten (41.7%) completed graduate education. Additional participant information is located in Table 2.

## Results

Four themes were conceptualized from participant interview data, including 1) personal experiences, opinions, and attitudes, 2) communities involved in breastfeeding, 3) public opinion and culture, and 4) breastfeeding alternatives. Themes and subthemes are elaborated through quotes labeled with participant ID and participant age (XXX, XX). Additional quotes are presented in Table 3.

### Personal experiences, opinions, and attitudes "Happy Mom, Happy Baby"

In the first theme, personal experiences, opinions, and attitudes, five subthemes were conceptualized, including 1) duration, 2) breastmilk production, 3) breastfeeding challenges, 4) benefits of breastfeeding, and 5) individualized experiences.

**Duration.** There was a wide range of participant breastfeeding durations, with some participants noting they breastfed for less than the recommended period. One participant stated she "heard a lot of women who tried and then after one month, they had to quit because they didn't have enough milk or had problems" (113, 59). Another participant explained, "Nowadays, more and more young women are not willing to breastfeed for a long time" (112, 58). One participant clarified why she stopped early and noted she "was tired" (119, 42). Another stated she "want[ed] to sleep train [her] baby" (110, 25). Each participant described an individualized experience.

Several participants indicated awareness of the recommended duration for breastfeeding. One participant stated, "Pediatricians say between six and nine months. Normally, mothers start to get some food at the age of six months, and they still breastfeed" (113, 59). Other participants described the pressure to adhere to the known guidelines despite personal

Table 2.  Participant Characteristics.

| Item | N (%) or M±SD |
|---|---|
| **Age (years)** | |
| Age of participants | 44.9 ± 11.1 (range = 25-62) |
| **Gender Identity** | |
| Cisgender woman | 24 (100) |
| **Relationship Status** | |
| Single | 2 (8.3) |
| One partner | 22 (91.7) |
| **Primary Language** | |
| Italian | 20 (83.3) |
| English | 4 (16.7) |
| Other | 1 (4.2) |
| **Pregnancies** | 48 Mean = 2; Mode = 2 |
| Vaginal birth | 23 (47.9) |
| Cesarean birth | 12 (25) |
| Miscarriage | 8 (16.7) |
| Abortion | 4 (8.3) |
| **Education** | |
| High school | 2 (8.3) |
| Undergraduate | 12 (50) |
| Graduate | 10 (41.7) |
| **Born** | |
| Italy | 18 (75) |
| United States | 2 (8.3) |
| Elsewhere | 4 (16.7) |
| **Household Income** | |
| Comfortable | 20 (83.3) |
| Just enough | 4 (16.7) |
| **Employment Status** | |
| Full-Time | 12 (50) |
| Part-Time | 8 (33.3) |
| Self-Employed | 3 (12.5) |
| Not Employed | 1 (4.2) |

Note: Data presented as Avg ± SD or n(%). Numbers that do not add to 100% reflect missing data. The age range is presented alongside the mean and standard deviation.

difficulties. One participant voiced, "it was super painful every single time… but I still exclusively breastfed for about six months" (116, 33). Additionally, one participant expressed her choice to breastfeed for the full recommended time, conveying, "my first child was breastfed, you're going to be shocked, until he was two" (113, 59). Many participants noted diverse reactions and experiences with these guidelines.

Alternatively, participants described breastfeeding experiences beyond the recommended duration. When explaining the motivation to continue beyond recommendations, one participant stated, "oh, I still have milk. I don't want to waste it" (110, 25). Some participants noted the effects of extended breastfeeding, with one participant commenting on her friend's breastfeeding experience: "[she] breastfe[d] until the baby was four and a half. Come on. Then you create a monster" (101, 37). Another participant noted, "my baby [is] six years [and] still

**Table 3. Additional Quotes.**

| Theme | Subtheme | Exemplar Quote |
|---|---|---|
| Personal Experiences, Opinions, and Attitudes | Duration | "It was kind of painful. And I remember talking to my mom and my mom was like, it was never painful. It was like always a really nice feeling. I was like, Oh, it was super painful every single time I felt my milk. But I still exclusively breastfed for about six months" (116, 33)<br>"I wanted to go until she was two. And she was just like one morning. She was just like, No" (116, 33)<br>"it can like about two years and a half that I finished that because I was like oh my god leave me alone" (107, 36)<br>"Oh my God a long time" (117, 40) |
| | Breastmilk Production | "And also, I feel I don't know why. But I, I feel like a lot of people didn't have a great milk supply" (116, 33) |
| | Issues with Breastfeeding | "It was a very, very ((pause)) stressful process, ((pause)) you are like fighting with your body, you feel like you are a cow, ((laughter)) and it's true!" (101, 37)<br>"I was almost anxious to be able to give him food" (122, 56)<br>"So actually, the breastfeeding totally threw me. Everything kind of threw me because I thought it was gonna be so much easier. I was so naïve" (116, 33)<br>"It's a lot. You don't you don't sleep. [And I didn't slept for nearly, I didn't slept for nearly for nearly three years" (120, 55)<br>"Other women who had children, have had children, who were telling me and judging me and telling me 'hey, It is not okay'. You know [inaudible], I was like, I my nipples hurt. They're bleeding and they we're like, 'it's nothing you'll get over it'. It was I don't know.'" (119, 42) |
| | Benefits of Breastfeeding | "It's the best option for the child, it's the best option for the mother because you don't have to stay to prepare the milk" (107, 36) |
| | Individualized Experiences | "You know, it's a mother's choice (122, 56)<br>"Yeah, you know, because we're all trying our best so it's whatever works for you (115, 38) |
| Public Opinion and Culture | Taboos Regarding Covering | "Yeah. Because there is no problem. In the south of Italy you, there is a sort of attention to this situation. So you have to cover her a little bit but I had no problem" (114, 57)<br>"And for me I just like, I just like I found a way to like cover for the most part" (105, 42)<br>"the men, for example, don't stare at you" (105, 42) |
| Breastfeeding Alternatives | Formula | "Hmm actually I bought the most convenient in cost because uh I found out that all these brands have the same formula, the same like composition. But some of them are more expensive because they invest more in advertising, but the formula is the same. So, I choose the cheapest one" (102, 32)<br>"Her milk for month is 100 no, it's around three 400 euros" (123, 29)<br>"Formula is better" (101, 37)<br>"I went to the hospital and the obstetrician um she um told me that it was better if I introduced the powder milk" (102, 32)<br>"No it wasn't, it was not allowed. Formula was not allowed. I was like I asked for it. I asked 'can I have?'" (119, 42)<br>"So for example, they gave her umm a bottle which was was not allowed in the other ward" (119, 42)<br>"The pediatric ward because she needed to get food that I couldn't provide and umm whatever was not allowed in the OBGYN ward umm was allowed there" (119, 42)<br>``I heard a lot of people that were struggling with the breastfeeding and so they help you, but at the same time, you're not discouraged to have the formula. So that's important I think. (124, 46) |
| | Breast Pumps | "So at the end, I didn't even have like enough milk because you have to pump like every two hours or so. And it didn't have time with her so it was less and less and less milk, so then I said okay that's it ((laughter)). And she's happy with formula, so, it's fine. (123, 29)<br>"Because at that time, I'm not pumping anymore and I had no milk. My milk is not producing a lot before because it depends on how you pump." (110, 25)<br>"they can also rent you some of them," (124, 46) |
| | Milk Banks | "Yea, it's very good. So, as I said, he got most of his milk through the milk bank." (118, 54) |

searching to touch my boobs because I was feeding him for about two years" (107, 36). Participants held varying opinions and experiences regarding breastfeeding for longer than the recommended duration.

**Breastmilk production.** Experiences with breast milk production strongly influenced participants' opinions surrounding breastfeeding. Individuals confident they produced sufficient breast milk had more favorable opinions surrounding breastfeeding. One participant detailed her positive experience, "With my kid, no problem with [him] hungry anytime and a lot of milk." (124, 46). Conversely, one participant mentioned a more negative

experience, "my experience left me with the feeling that I didn't have enough for him" (122, 58). Other participants expressed strong emotional responses stemming from the need to supplement with formula, "we had to resort to the bottle, and that just killed me" (118, 54). Many participants experienced emotional turmoil when they could not produce enough breast milk. Overall, participants' personal milk production amount led to strong emotional responses and shaped participants' opinions on breastfeeding.

**Breastfeeding challenges.** Many participants noted issues and overwhelming feelings within their personal breastfeeding experiences. Participants expressed a lack of support regarding the onset of their breast milk. One participant explained, "I did not know at first that, but I had a lot of milk. It was just difficult because I needed to do [breastfeeding] all by myself" (110, 25). Another participant echoed these attitudes, stating, "When you leave the hospital, I had masses of milk, and no one told me, and I didn't think about it, the milk, we were just waiting to feed him" (118, 54). Others described their feelings when faced with breastfeeding issues, one participant admitting, "I felt guilty about it. I felt that I wasn't ready for my daughter. It was heartbreaking" (119, 42). Another participant stated, "then mastitis came over. I first felt disappointed. I cannot even do this" (113, 59). Additionally, some participants mentioned the exhaustion they felt, with one explaining, "I cannot leave, and I am breastfeeding. I am always exhausted with everything, I am exhausted, I'm not even able to have a shower" (124, 46). Another shared, "[Breastfeeding] was very stressful. I felt like a cow. Like the first three months w[ere] very intense" (120, 55). Participants identified issues with breastfeeding that led to heightened emotions.

Participants also mentioned issues of nipple tenderness and latching pain. One participant described, "I almost quit. I was like this [is] too much, too painful" (115, 38). Another participant explained, "breastfeeding is hard. It's hard work….It takes a lot of time to get your babies [to] latch properly… for me, it was painful" (119, 42). The same participant also noted, "my nipples were cracked and bleedy and so [my daughter] would suck my blood and she would throw up because of that. Yeah, it hurt me. It was painful for me. It was painful for her" (119, 42). Nipple soreness and latching pain are physical changes many participants identified through their personal experiences with breastfeeding.

**Benefits of breastfeeding.** A majority of participants believed breastfeeding was superior to other alternatives. Several participants explained the benefits of their relationship with their baby: "It's a nice bonding…I think the children feel really loved" (120, 55) and "breastfeeding is beautiful. There's a very great connection with [the baby]" (101, 37). Participants also described the health benefits of breastmilk. One participant stated, "just because I had the milk, I gave protection to the small baby" (114, 57), and another affirmed this: "in breastfeeding, you are giving your kids antibodies…the support of the immune system" (124, 46). Overall, participants noted breastfeeding as the preferred option, citing the immune system benefits and stronger parent-child connection.

**Individualized experiences.** Participants described individualized feeding regimens to fulfill the needs of each mother and baby. One participant shared, "it really depends on the situation you are in. I think most of the women would do it if they can" (120, 55). Another participant agreed by saying, "I think that whatever is good for them. It's okay" (109, 42). Several participants similarly believed feeding methods should provide nutrients to the baby in the most comfortable process for the mother and baby. One participant said, "it doesn't make any less of a mom… It's okay if even if you're not breastfeeding. Happy mom, happy baby" (110, 25). Another participant echoed this by saying, "I think a fed baby is a happy baby" (115, 38). Participants expressed breastfeeding is an overwhelming and stressful experience, and ultimately, a mother's decision to breastfeed is unique to their situation.

## Communities involved in breastfeeding "They have to encourage and support women"

In the second theme, communities involved in breastfeeding, three subthemes were conceptualized, including 1) diverse communities, 2) diverse breastfeeding education, and 3) workplace support.

**Diverse Communities.** While breastfeeding, participants interacted with multiple communities, including but not limited to, healthcare providers, midwives, support groups, other moms, and family. These communities provided both education and support through participants' breastfeeding processes. Participants expressed varied attitudes toward these communities regarding breastfeeding. Some were positive,

I had a private midwife who came and she told me about breastfeeding lying down, which I never managed to do with my other two, partly because I had so much milk, but she helped me learn how to do that. And it was just the best (104, 32).

While others had a negative experience: "[the nurses] are very rude. They come to you, 'sponge your boob, come on you have to breastfeed'" (101, 37). Another participant noted, "they wouldn't help me– I was like, I need help, my child is not coping well" (119, 42). This participant continued, "it was a long process. It wasn't easy at all, but nobody really helped me in the OBGYN" (119, 42). Communities directly involved with participants' breastfeeding influenced their perception of their experience.

**Diverse breastfeeding education.** In terms of education, participants also had both positive and negative experiences. One participant mentioned YouTube pre-birth courses as "useless," noting, "you have to teach a woman, to really teach about breastfeeding, about all the problems" (101, 37). Another participant builds on this by mentioning a lack of information, stating, "I found that it's hard, and information is not volunteered to you here…you have to ask a specific question, and you have to also ask it to the right person" (104, 32). Other participants disagreed and felt ample resources were available to those who wished to utilize them. One mentioned a positive personal experience with a midwife,

It's awesome. And I feel like there's a lot of resources for helping with breastfeeding. I remember when I was having trouble with latching… I had a one-on-one with a midwife through the public hospital. All I did was an appointment just to check to make sure that the breastfeeding was going okay. And there were a lot of people who took advantage of that resource (116, 33).

One participant echoed this positive sentiment, stating how the healthcare system has programs to support breastfeeding,

So, they have to encourage women and support women with breastfeeding. I think that the program works, like they got this…certificate. And they got money with it if they convinced women and they show that they're doing those things, you know, like 'Yeah, we're pushing breastfeeding' (119, 42).

Other participants discussed support groups that aided their efforts with similar programs, "so there is an international league called 'Leche League,' so they help women for breastfeeding and so I went to meetings with this league, and I learned things that were very helpful for breastfeeding" (121, 58). Support groups like this one provided both support and education.

**Workplace support.** Regarding breastfeeding in the workplace, one participant noted, "it's totally accepted. It's totally fine. If you work for a big company, they'll give you additional breaks for you to breastfeed" (116, 33). Another participant shared, "When I took the bar

exam, it took a lot of hours, and this woman there had to leave to go breastfeed, but they just gave her a corner to go pump" (117, 40). Aside from the healthcare and workplace community, participants utilized breastfeeding support from familial and personal connections. One participant described her experiences with her family during breastfeeding,

My mom was here with me. And she had never breastfed herself and she was just helpful in terms of just support, like emotional support. So, I just reached out to cousins, had a lot of cousins with babies and ones that breastfed, so just kind of asked them about it. After I left the hospital, I didn't talk to any other professionals, no lactation consultants or anybody. So that support was just enough for it all (115, 38).

The same participant built on this further by explaining her experience of reaching out to personal maternal connections: "you're reaching out to every woman who's breastfe[d], and you say, 'What was your experience? Is this normal?'" (115, 38). Overall, participants chose to utilize various communities through breastfeeding. Each community uniquely aided or failed to provide education and support to each participant.

## Public opinion and culture "If you don't breastfeed, you are not a good mother"

In the third theme, public opinion and culture, three subthemes were conceptualized, including 1) breastfeeding in public, 2) taboos regarding covering, and 3) societal pressures.

**Breastfeeding in public.** Participants described various factors involved in public breastfeeding concerning privacy and cultural norms. When asked whether there are private spaces to breastfeed in public, one participant stated, "I don't think there are locations designated for breastfeeding, but I would say that I usually try to find more private places, even if I was in a public space maybe not going in the middle of kids playing" (121, 51). Another participant echoed this sentiment, explaining, "there's little places you can go for breastfeeding." (115, 38). Some participants indicated they felt comfortable breastfeeding among others. One participant noted, "I felt comfortable breastfeeding with other Italian moms around because I've seen them do it, and they seemed very relaxed about it" (104, 32). Other participants agreed; "I never felt that people were looking at me in a bad way because I was breastfeeding" (121, 51) and "you can do it everywhere" (102, 32). One participant noted a differing personal opinion, stating, "it was not taboo at my time, but I really didn't feel very comfortable, so I breastfed particularly in place that was private and not exposed" (122, 56) and "maybe I don't want to see you breastfeeding your child, you know?" (122, 56). Generally, participants exhibited a wide range of opinions and viewpoints on the culture of public breastfeeding.

**Taboos regarding covering.** Participants expressed various feelings about covering while breastfeeding in public. One participant noted society expects covering, stating, "[in the] Catholic religion….everything must be hidden" (112, 58). Another participant reiterated the importance of covering in specific places, explaining, "really you don't [breastfeed] in a church or in a special place" (114, 57) but also stated, "if you are in the garden, you don't have to cover" (114, 57). One participant voiced her belief that the culture around covering is changing as she explained, "Sometimes they cover themselves, but I think it's something that they did maybe 20 30 years ago. Now, our generations don't do that anymore" (123, 29). Many participants expressed different views on covering.

**Societal pressures.** Many participants expressed public opinion and culture within the community puts pressure on mothers to breastfeed. One participant said, "I felt bad because in this culture, they feel this stereotype that if you don't breastfeed, you are not a good

mother" (102, 32). Participants voiced community opinions towards breastfeeding placed pressure and judgment on a mother's feeding abilities, with one participant stating,

> Yeah, it was a really brutal experience for me… I felt like it was my fault because they were judging me. Like 'you're not maternal enough'. 'You're not giving your daughter enough attention and enough warmth and enough love to, you know, survive'. I was feeling like shit. I remember I cried every time and constantly. And they wouldn't let me sleep. That's one thing that I found (119, 42).

Additionally, participants expressed their community places pressure on the need for women to breastfeed, so much so that mothers may change their desired feeding practices to meet cultural standards. One participant stated, "Not everybody is meant to do it. I had friends who really didn't want to do it but felt pressured" (101, 37). Participants identified the public's opinion towards breastfeeding may pressure a mother to breastfeed. Furthermore, women may face negative public opinions by not conforming to these cultural standards.

### Breastfeeding alternatives "All of my friends supplement"

In the fourth theme, breastfeeding alternatives, three subthemes were conceptualized, including 1) formula, 2) breast pumps, and 3) milk banks.

**Formula.** Formula is a common alternative many participants mentioned using either as a supplement to breastfeeding or as a replacement entirely. Participants discussed the differences between breastfeeding and formula. One participant mentioned the only significant difference was antibodies given through breastfeeding, "that is what is different from the formula, not in terms of nutrition, but just in terms of the support of the immune system" (124, 46). However, some participants considered breastfeeding and formula as interchangeable. One mentioned, "I think formula and breastfeeding are equally important, and I think that regardless of how the baby is fed…. there's not one that's better than the other" (115, 38). While acknowledging the differences between formula and breastfeeding, most participants still had positive experiences with formula.

Additionally, many participants thought having an alternative to breastfeeding and the pump was important. One participant positively mentioned this, "no! We are not meant to do this. If I don't care to pump, then if I don't feel happy in breastfeeding. Formula!" (101, 37). Some participants detailed using formula to supplement breastfeeding. One participant explained the possibility of using a formula to supplement the lack of milk production, stating, "But if you cannot produce, like, if it is not enough for your baby, powdered milk, it's okay" (110, 25). Another participant mentioned feeling more secure as a result of using formula,

> In the evening, I was giving him formula also because it help him to sleep better and I was feeling more secure because you didn't know… you never know how much milk from a human. So, formula then you know (101, 37).

Another participant said, "all of my friends supplement." (116, 33). Supplementation through formula was a common topic among participants.

Some participants used formula out of necessity. One participant shared, "I really wanted to breastfeed, but unfortunately, after one month, I introduced artificial milk because I didn't have much milk" (102, 32). Another participant explained, "obviously, [my daughter] needed milk, I wasn't giving her much, and she wasn't putting on weight" (118, 54). One participant used formula because her baby struggled to latch,

I didn't… it was very hard because I have big breasts, and she was so little toward the beginning. And she couldn't, like, she was latching, and then letting go, latching, and letting go. And she wasn't eating. So she was screaming the whole time. And we were very stressed about that. So we just switch to formula very fast (123, 29).

Formula is commonly used as both a supplement and an alternative to breastfeeding for a variety of reasons.

**Breast pumps.** In terms of acquiring breast pumps, many participants purchased them. One participant stated, "I had a breast pump… I bought it before my son was born" (121, 51). Another participant described an option to rent but explained she felt uncomfortable renting, "I wouldn't feel comfortable in that. That's really personal. I think they cannot give you just the machine, they give you everything" (124, 46). One participant who was lent a pump noted, "I actually borrowed it from another mum from Florence expat mom's group, she lent it to me. But I did look at other options before that. I looked at buying one" (104, 32). The decision whether to rent or purchase a pump is personal, so some participants were more comfortable using a breast pump than others.

Several participants described success with their pumps, "my pump was very useful" (121, 51). Another noted, "I had to grab the pump. And I was working, and after three hours, your boobs are exploding. It's stressful if a woman has to go back to work" (101, 37). One participant described pumping out of necessity, stating, "I had to sleep on a one-hour cycle like every three hours and then just to be able to pump milk for her and to bring it down to her" (105, 42). While another participant pumped for convenience, explaining, "I would pump into a bottle sometimes for a night feed for her dad, but other than that, I've just breastfed" (115, 38). Several participants described positive experiences with pumping.

Some participants disliked their pumps because of pain or a lack of milk production. One participant noted, "I tried this pump because I said, 'Probably I have too much milk.' But no, it was not something that I liked" (113, 59). While another participant noted the pain that came with pumping, "they had to manually pump me. It was the most painful thing I've ever been through except for birth. So painful... I have tiny boobs, so it was really, really painful" (116, 33). Another participant discussed how pumping did not work for her, stating, "in the next few weeks, we tried, I tried to do things like pump, stuff like that to build up my milk, but it didn't work, so we had to switch" (118, 54). Each participant had a different experience; some could breastfeed without pumping, but others found it helpful.

**Milk banks.** Milk banks are an alternative to breastfeeding that still allows mothers to feed their children with breast milk, in case they cannot breastfeed their child. Many participants used milk banks to donate or acquire breast milk for their children. One participant explained her excess of milk led her to donate, "I had a lot of milk. So, I give my milk to the milk bank" (114, 57). Another participant shared a similar experience,

I would say that actually, I was also able to donate some of my milk because I had a lot. So, there's this milk bank here because we have the pediatric hospital that is really, really big here. So actually, you can donate for people that [are] not able to (124, 46).

Other participants needed milk for their children, which they received from milk banks. One shared, "luckily, the hospital had a milk bank, so he would get milk from other mothers" (118, 54). Overall, milk banks seem to be a reliable source for obtaining and donating breast milk.

## Discussion

Researchers completed 24 in-depth interviews with women living in or around Florence, Italy, to further understand breastfeeding behaviors, personal attitudes, and cultural opinions. Study

results identified four main themes from participant interviews: 1) personal experiences, opinions, and attitudes towards breastfeeding, 2) communities involved in breastfeeding support, 3) public opinion and cultural factors affecting breastfeeding, and 4) breastfeeding alternatives. Participants had varying breastfeeding durations, influenced by personal experiences and external pressures. Breast milk amount shaped opinions and emotions about breastfeeding, impacting participants' attitudes. Breastfeeding issues such as pain, exhaustion, and latch difficulties were common, with participants expressing both positive and negative feelings. The benefits of breastfeeding mentioned included bonding and health advantages for the baby. Communities, including healthcare providers, midwives, support groups, and family, played various roles in education and support during breastfeeding. Public opinion and culture exerted pressure on mothers to breastfeed, and opinions on breastfeeding in public and covering varied. Breastfeeding alternatives, such as formula, breast pumps, and milk banks, were utilized for various reasons, influenced by personal circumstances and preferences.

Participants expressed breastfeeding is a personal experience dependent on each mother's situation. Many possessed knowledge of the World Health Organization (WHO) recommended duration, consistent with past research, suggesting Italy promotes breastfeeding behaviors [14]. Even with this knowledge, some participants noted a stigma surrounding extended breastfeeding until age two, implying no widespread adherence to known guidelines. This adds to past research that reported low global adherence to exclusive breastfeeding until the age of 6 months [18]. Further, participants emphasized intense feelings of guilt, anxiety, pain, and exhaustion when faced with breastfeeding struggles, specifically influenced by high or low milk production amounts. These results align with past research that shows a correlation between production amount and anxiety [19]. Many participants reported these issues caused struggles in reaching the recommended breastfeeding duration. Despite these struggles, participants also described positive feelings of enhanced wellness and connection with their baby derived from breastfeeding, expanding upon literature showing the benefits of breastfeeding [14].

Participants benefited from communities that offered personalized education with a strong emphasis on patience and support throughout the breastfeeding process. Many women preferred to learn breastfeeding techniques at their own pace and believed healthcare workers should avoid a rushed or forced approach. This finding highlighted the successful approaches healthcare professionals take in breastfeeding education. This builds on past research that showed antepartum and comprehensive breastfeeding education are positively associated with longer durations of exclusive breastfeeding [20,21]. Postnatal breastfeeding education has similarly been shown to enhance mothers' breastfeeding knowledge, confidence, and success rates, thereby supporting sustained breastfeeding practices [22]. Resources regarding the breastfeeding process vary in both media and content. Still, many participants identified support communities as a key factor in helping participants find relevant educational materials and support for their unique needs. Participants noted social platforms such as YouTube to be largely unhelpful, whereas one-on-one in-person appointments were more effective. Beyond healthcare workers, supportive communities for participants included support groups, other moms, and families. Results are consistent with previous research that shows access to community resources positively impacts the incidence of breastfeeding practices [23]. Communities directly involved with the participants' breastfeeding journeys influenced their perception of their experiences. Workplaces have supported participants by giving postpartum women ample time off work to breastfeed. This finding adds to the existing literature that parental leave provisions are meant to improve the mother and baby's health and welfare [14]. Despite these workplace efforts, participants more frequently identified personal maternal connections and family support as the most significant community contributing to support through their breastfeeding journey.

Culture widely influences breastfeeding in various ways, especially in public environments. Many participants did not mention the availability of designated private spaces for breastfeeding. However, many noted they would often prefer areas with a higher prevalence of mothers (e.g., parks) or semi-private non-designated spaces. Some participants felt uncomfortable breastfeeding in public, leading to many seeking these more private areas. This was consistent with another study that emphasized how Italian mothers specifically were less likely to breastfeed in public as a result of cultural views [10]. Globally, mothers' public breastfeeding decisions are often shaped by their sociocultural environment, with many opting to seek private spaces or "cover up" to align with social norms and expectations [24,25]. These results suggest mothers may be more open and/or comfortable breastfeeding if specific designated breastfeeding spaces were available and normalized. Additionally, many participants felt the need to cover themselves in some way (e.g., scarves and shawls) while breastfeeding in public as a result of public opinion and religious influence, specifically Catholicism. This need to cover was accentuated in 'special places' such as churches, further showing religious influence. This builds on previous research indicating that societal disapproval of public breastfeeding is particularly evident in specific settings and circumstances, such as during church services [26]. However, some participants believe this culture is changing as new generations become accustomed to public breastfeeding.

Overall, public opinion in Italy pushes mothers to breastfeed. Many mothers received negative maternal judgment when choosing not to breastfeed. Additionally, the public opinion stated one may be a bad mother if one chooses not to breastfeed. This agreed with other studies that found this same harsh public opinion surrounding breastfeeding [14]. This pressure was so deep and ingrained that participants mentioned mothers choosing to breastfeed only to meet cultural standards and not because they wanted to. Public opinion is contrasting in this way; while some wish to decrease breastfeeding in public spaces (if not covered or in a semi-private area), there is also a large overall push to breastfeed. In earlier studies, a similar paradox was seen: breastfeeding is natural but still slightly stigmatized publicly [14]. This is contradictory because the public wants to increase breastfeeding but does not wish to see it.

As mentioned, formula is a breastfeeding alternative primarily used as a supplement or replacement for breastfeeding. Many participants described needing formula due to a lack of milk production, but others used formula to supplement breastmilk out of convenience or both. Previous literature suggests pain during lactation or lack of milk production leads to "a higher risk of exclusive breastfeeding discontinuation," with some participants sharing this experience [3]. Despite breastfeeding being the 'most natural' infant feeding method, most participants did not see health differences between formula and breastmilk, aside from breast milk providing antibodies and protection from illness [6,8] Overall, this revealed women generally do not see formula as less effective or nutritional than breast milk. Regarding breast pumps, some participants loved their pumps, while others had painful or unsuccessful experiences. Furthermore, milk banks were helpful to mothers who needed breastmilk or those who produced large amounts and could donate. In conclusion, alternative breastfeeding methods were viewed as acceptable and sometimes preferred.

## Implications

The implications of this research extend across various societal, political, workplace, and health dimensions, underscoring the necessity of addressing diverse cultural attitudes and public perceptions surrounding breastfeeding. Efforts should focus on normalizing breastfeeding through public education campaigns that dispel myths and reduce societal pressure on mothers. Designated breastfeeding spaces in public areas could provide a compromise, accommodating mothers who feel uncomfortable in fully public spaces. Additionally,

understanding and respecting individual preferences regarding covering while breastfeeding in public is essential for creating a culturally sensitive environment. Politically, the study emphasizes the importance of comprehensive workplace policies that support breastfeeding mothers, including provisions for breast pumping. Policymakers should consider the societal pressure on mothers to breastfeed and work towards initiatives that challenge and reshape these cultural norms. Advocacy for inclusive policies, public health campaigns, and educational programs can provide a more supportive environment for mothers who choose alternative feeding methods.

On the workplace front, this research underscores the significance of accommodations for breastfeeding mothers, emphasizing the need for a positive and supportive professional environment. Workplace initiatives that encourage and normalize breastfeeding-friendly practices, such as additional breaks for breastfeeding, play a vital role in fostering a healthy work-life balance for new mothers. Regarding health implications, the study highlights the need for comprehensive breastfeeding education and support within healthcare systems. Beyond providing accurate information, healthcare providers should prioritize patient-paced education, aligning with participants' preferences for personalized learning. The emotional impact of breastfeeding challenges, such as insufficient milk production and painful experiences, emphasizes the need for mental health support integrated into postpartum care. The study reinforces the importance of support networks, including healthcare providers, support groups, and family, in shaping positive breastfeeding experiences and enhancing overall maternal and infant health outcomes.

## Strengths, limitations, and future research

This study's comprehensive approach, addressing pregnancy, postpartum experiences, and breastfeeding, provided a well-rounded view of women's perspectives within Florence's unique cultural context. Led by a leading global researcher in family planning with extensive experience conducting research in Italy, the study leveraged a team of 24 trained, all-female undergraduate interviewers who conducted interviews over two months. This extended presence allowed researchers to immerse themselves in the community, fostering participant comfort with sensitive topics. The all-female team likely contributed to an open environment, encouraging candid discussions about personal subjects like breastfeeding and menstruation. Although a large, varied team may have introduced slight inconsistencies in data collection, the diverse perspectives enriched cultural relevance and captured a broader array of insights.

Despite these strengths, certain limitations may have affected the study's generalizability. Conversational English proficiency was required for participation, which could have narrowed the diversity of backgrounds and experiences represented, as English-speaking participants may have shared similar educational or cultural values. Additionally, the sample primarily consisted of older, educated, cisgender women from higher socioeconomic backgrounds, reflecting Florence's demographic trends but limiting diversity across gender, sexual orientation, and socioeconomic status. While these factors may reduce the applicability of findings to other regions or populations, they provide valuable insights into a specific demographic. Future studies could address these limitations by broadening participant demographics, exploring varied cultural contexts within Italy, and employing longitudinal designs to better capture how attitudes and experiences evolve over time.

## Conclusions

This research revealed the complex dynamics that shaped breastfeeding experiences in Florence, where personal motivations, societal pressures, and cultural expectations intersected. Each mother's

breastfeeding journey was unique, influenced by individual goals and the sometimes-conflicting messages from her community. Healthcare providers, midwives, and support groups offered encouragement and obstacles, impacting each mother's perception and decisions. Public opinion and cultural norms strongly affected comfort with breastfeeding in public, as well as the pressure to meet idealized standards. Yet, alternatives like formula, breast pumps, and milk banks provided vital options, addressing diverse challenges and reinforcing the need for accessible choices. Ultimately, this study highlights the importance of ongoing dialogue, supportive policies, and tailored resources to create a more empowering environment for breastfeeding, respecting each mother's path.

## Supporting Information

**S1 Questionnaire. Inclusivity in Global Research Questionnaire.**
(DOCX)

## Acknowledgments

We thank the students who participated in the summer 2023 Purdue University Investigating Women's Reproductive and Sexual Health Issues in Florence, Italy, study abroad program for supporting data collection, transcription, and overall collaboration on the project. We also thank our Florence University of the Arts colleagues for their partnership and project support.

## Author contributions

**Conceptualization:** Andrea L. DeMaria.

**Data curation:** Megan Morley.

**Formal analysis:** Megan Morley, Anjali Natarajan.

**Funding acquisition:** Andrea L. DeMaria.

**Investigation:** Megan Morley, Anjali Natarajan.

**Methodology:** Andrea L. DeMaria.

**Project administration:** Andrea L. DeMaria.

**Resources:** Andrea L. DeMaria.

**Supervision:** Nicole A. Stepp, Andrea L. DeMaria.

**Visualization:** Andrea L. DeMaria.

**Writing – original draft:** Megan Morley, Anjali Natarajan.

**Writing – review & editing:** Megan Morley, Anjali Natarajan, Nicole A. Stepp, Andrea L. DeMaria.

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
