## [Decision Letter · Decision Letter 0]

4 Oct 2024

PGPH-D-24-00358

“It’s a mother’s choice”: Exploring Personal Experiences, Community Support, Cultural Influences, and Breastfeeding Alternatives in Florence, Italy

Dear Dr. Morley,

Thank you for submitting your manuscript to PLOS Global Public Health. After careful consideration, we feel that it has merit but does not fully meet PLOS Global Public Health’s publication criteria as it currently stands. Therefore, we invite you to submit a revised version of the manuscript that addresses the points raised during the review process.

Please carefully revise your manuscript in line with the reviewer comments and suggestions to improve your manuscript.

We look forward to receiving your revised manuscript.

Kind regards,

Jennifer Tucker, PhD

Staff Editor

Journal Requirements:

**Please only choose the relevant sentences from below**

1. Please clarify all sources of funding (financial or material support) for your study. List the grants (with grant number) or organizations (with url) that supported your study, including funding received from your institution. 

2. State the initials, alongside each funding source, of each author to receive each grant.

3. State what role the funders took in the study. If the funders had no role in your study, please state: “The funders had no role in study design, data collection and analysis, decision to publish, or preparation of the manuscript.”

4. If any authors received a salary from any of your funders, please state which authors and which funders.

3. In the online submission form, you indicated that "The data and corresponding codebook are available on request.". 

3. Uploaded as supplementary information.

4. Please insert an Ethics Statement at the beginning of your Methods section, under a subheading 'Ethics Statement'. It must include:

1) The name(s) of the Institutional Review Board(s) or Ethics Committee(s)

2) The approval number(s), or a statement that approval was granted by the named board(s) 

3) (for human participants/donors) - A statement that formal consent was obtained (must state whether verbal/written) OR the reason consent was not obtained (e.g. anonymity). NOTE: If child participants, the statement must declare that formal consent was obtained from the parent/guardian.

Additional Editor Comments (if provided):

Reviewers' comments:

Reviewer's Responses to Questions

**Comments to the Author**

1. Does this manuscript meet PLOS Global Public Health’s publication criteria ? Is the manuscript technically sound, and do the data support the conclusions? The manuscript must describe methodologically and ethically rigorous research with conclusions that are appropriately drawn based on the data presented.

Reviewer #1: Partly

Reviewer #2: Yes

2. Has the statistical analysis been performed appropriately and rigorously?

Reviewer #1: Yes

Reviewer #2: N/A

3. Have the authors made all data underlying the findings in their manuscript fully available (please refer to the Data Availability Statement at the start of the manuscript PDF file)?

Reviewer #1: Yes

Reviewer #2: Yes

4. Is the manuscript presented in an intelligible fashion and written in standard English?

Reviewer #1: Yes

Reviewer #2: Yes

5. Review Comments to the Author

Reviewer #1: Thank you for the opportunity to review this manuscript entitled “It’s a mother’s choice”: Exploring Personal Experiences, Community Support, Cultural Influences, and Breastfeeding Alternatives in Florence, Italy.

The detailed response in provided in an attached document.

In summary the recommended amendments relate to:

• Including up to date contextual literature

• Providing a methodological rationale for the choice of the study site

• Some amendment of theme and sub-theme headings in results

• Strengthen the discussion by including up to date citations

• Avoid overstating the strengths of the study and properly acknowledge methodological limitations

Reviewer #2: Thank you for the opportunity of reviewing this important manuscript. Please consider the suggested recommendations to make the manuscript more robust. The comments are directed to sections of the abstract, introduction, methods and results.

6. PLOS authors have the option to publish the peer review history of their article (what does this mean? ). If published, this will include your full peer review and any attached files.

**Do you want your identity to be public for this peer review?** For information about this choice, including consent withdrawal, please see our Privacy Policy .

Reviewer #1: **Yes: ** Dr Helen Mulcahy

Reviewer #2: **Yes: ** Dr. Adwoa Gyamfi

---

## [Decision Letter · Decision Letter 1]

24 Jan 2025

“It’s a mother’s choice”: Exploring Personal Experiences, Community Support, Cultural Influences, and Breastfeeding Alternatives in Florence, Italy

PGPH-D-24-00358R1

Dear Ms. Morley,

We are pleased to inform you that your manuscript '“It’s a mother’s choice”: Exploring Personal Experiences, Community Support, Cultural Influences, and Breastfeeding Alternatives in Florence, Italy' has been provisionally accepted for publication in PLOS Global Public Health.

Best regards,

Julia Robinson

Executive Editor

Reviewer Comments (if any, and for reference):

Reviewer's Responses to Questions

**Comments to the Author**

1. If the authors have adequately addressed your comments raised in a previous round of review and you feel that this manuscript is now acceptable for publication, you may indicate that here to bypass the “Comments to the Author” section, enter your conflict of interest statement in the “Confidential to Editor” section, and submit your "Accept" recommendation.

Reviewer #1: All comments have been addressed

2. Does this manuscript meet PLOS Global Public Health’s publication criteria ? Is the manuscript technically sound, and do the data support the conclusions? The manuscript must describe methodologically and ethically rigorous research with conclusions that are appropriately drawn based on the data presented.

Reviewer #1: Yes

3. Has the statistical analysis been performed appropriately and rigorously?

Reviewer #1: N/A

4. Have the authors made all data underlying the findings in their manuscript fully available (please refer to the Data Availability Statement at the start of the manuscript PDF file)?

Reviewer #1: Yes

5. Is the manuscript presented in an intelligible fashion and written in standard English?

Reviewer #1: Yes

6. Review Comments to the Author

Reviewer #1: I have reviewed the response to all reviewer comments and consider that the revised manuscript provided has addressed all of these.

I wish the authors well.

7. PLOS authors have the option to publish the peer review history of their article (what does this mean? ). If published, this will include your full peer review and any attached files.

**Do you want your identity to be public for this peer review?** For information about this choice, including consent withdrawal, please see our Privacy Policy .

Reviewer #1: **Yes: ** Dr Helen Mulcahy
